# CMGC Kinases in Health and Cancer

**DOI:** 10.3390/cancers15153838

**Published:** 2023-07-28

**Authors:** Iftekhar Chowdhury, Giovanna Dashi, Salla Keskitalo

**Affiliations:** 1Institute of Biotechnology, University of Helsinki, 00014 Helsinki, Finland; iftekhar.chowdhury@helsinki.fi (I.C.);; 2Helsinki Institute of Life Science, University of Helsinki, 00014 Helsinki, Finland

**Keywords:** CMGC kinases, cancer, protein–protein interaction, therapy

## Abstract

**Simple Summary:**

CMGC kinases, named after the initials of its main kinase families, CDKs, MAPKs, GSKs, and CLKs, play critical roles in many cellular processes. Dysregulation of these kinases has been associated with cancer development and progression, highlighting their significance in cancer biology. The success of certain kinase inhibitors demonstrated the therapeutic potential of targeting CMGC kinases. However, the clinical application of CMGC kinase inhibitors is not without its challenges. Here, we discuss the challenges including resistance; off-target effects; patient stratification, which can be addressed through the development of next-generation inhibitors; combination therapies; and exploiting protein–protein interactions involving CMGC kinases. Understanding these interactions could reveal novel targets and allow for the design of drugs that disrupt these specific interactions, offering another layer of precision in targeting these key players in cancer biology. Through continued research and development, next-generation inhibitors and novel therapeutic strategies could play a pivotal role in improving outcomes for cancer patients.

**Abstract:**

CMGC kinases, encompassing cyclin-dependent kinases (CDKs), mitogen-activated protein kinases (MAPKs), glycogen synthase kinases (GSKs), and CDC-like kinases (CLKs), play pivotal roles in cellular signaling pathways, including cell cycle regulation, proliferation, differentiation, apoptosis, and gene expression regulation. The dysregulation and aberrant activation of these kinases have been implicated in cancer development and progression, making them attractive therapeutic targets. In recent years, kinase inhibitors targeting CMGC kinases, such as CDK4/6 inhibitors and BRAF/MEK inhibitors, have demonstrated clinical success in treating specific cancer types. However, challenges remain, including resistance to kinase inhibitors, off-target effects, and the need for better patient stratification. This review provides a comprehensive overview of the importance of CMGC kinases in cancer biology, their involvement in cellular signaling pathways, protein–protein interactions, and the current state of kinase inhibitors targeting these kinases. Furthermore, we discuss the challenges and future perspectives in targeting CMGC kinases for cancer therapy, including potential strategies to overcome resistance, the development of more selective inhibitors, and novel therapeutic approaches, such as targeting protein–protein interactions, exploiting synthetic lethality, and the evolution of omics in the study of the human kinome. As our understanding of the molecular mechanisms and protein–protein interactions involving CMGC kinases expands, so too will the opportunities for the development of more selective and effective therapeutic strategies for cancer treatment.

## 1. Introduction

Eukaryotic cells are characterized by the simultaneous activity of multiple molecular networks. Central to these networks are the reversible reactions of protein phosphorylation, catalyzed by protein kinases, and dephosphorylation, catalyzed by protein phosphatases. Protein kinases play a crucial role in signal transduction by phosphorylating specific protein substrates. Changes in the phosphorylation status of numerous proteins are closely associated with their activity, cellular location, and interactions with other proteins. Consequently, alterations in protein phosphorylation can influence a wide range of cellular processes, including metabolism, transcription, cell cycle progression, cytoskeletal rearrangement, cell motility, apoptosis, and differentiation. Dysregulation of protein phosphorylation has been implicated in various human diseases, including cancer. Phosphorylation involves the transfer of a phosphate group from an ATP molecule to a target protein’s serine (S), threonine (T), or tyrosine (Y) residues by protein kinases. Based on their target amino acid selectivity, protein kinases can be classified into serine/threonine (Ser-/Thr-specific), tyrosine (tyrosine-specific), and dual-specificity protein kinases, which primarily phosphorylate serine/threonine residues but can also target tyrosine residues. Serine/threonine protein kinases are the predominant regulators of most cellular processes, with phosphorylation event rates yielding ratios of 1800(Ser):200(Thr):1(Tyr) [1].

All eukaryotic protein kinases share a conserved catalytic core, comprising N- and C-terminal lobes with an ATP-binding cleft between them. The N-lobe contains five β-strands and an α-helix, while the C-lobe is primarily composed of helices [2]. The N-lobe β-strands serve as ATP-binding sites, and the α-helix acts as a regulatory unit crucial for kinase activation, connecting the N- and C-lobes. The C-lobe houses the substrate binding groove and the activation segment, which encompasses the catalytic loop, DFG motif, activation loop, P + 1 loop, and APE region [3]. The activation segment’s catalytic and regulatory functions facilitate the accurate transfer of the phosphate residue to the substrate. Mammalian protein kinases are tightly regulated, with the DFG-motif-regulated phosphorylation of the activation loop functioning as an activation/deactivation switch [2]. Activation loop phosphorylation represents the most common mechanism for regulating kinase activity [4]. Phosphorylated protein kinases are typically in an active ‘on’ state, while dephosphorylated kinases assume a basal ‘off’ state. The three-dimensional structure of the activation segment in different active protein kinases is highly conserved, whereas inactive conformations exhibit significant variation, allowing for kinase-specific activation mechanisms and high specificity [2]. Protein kinase activation can also be dependent on the priming phosphorylation of the substrate [3].

Manning and colleagues identified 518 protein kinases in the human genome in 2002, constituting one of the largest gene families in eukaryotes and accounting for approximately 2% of all human genes [5]. This collection of over 500 protein kinases, which forms the human ‘kinome’, was further categorized into seven major families based on structural similarities within their catalytic domains. Cyclin-dependent kinases (CDKs), mitogen-activated protein kinases (MAPKs), glycogen synthase kinases (GSK3s), CDC-like kinases (CLKs), and related kinase families were grouped together to form the CMGC kinase family based on similarities in their kinase domains [5]. The CMGC family is highly conserved across various species, from nematodes to humans, suggesting that their functions are essential for the survival of these organisms.

CMGC kinases possess a conserved kinase core, similar to other protein kinases. However, they also share a unique CMGC-insert segment absent in other protein kinases [3]. Located in the C-lobe, the CMGC-insert is involved in binding co-proteins that participate in kinase function [6]. This domain exhibits low homology among family members, thereby enabling substrate specificity. CMGC kinases are primarily regulated through tyrosine phosphorylation in the activation loop [7] or pre-phosphorylation of the substrate, which prepares it for sequential recognition and phosphorylation by these kinases [8]. Another shared functional feature is their preference for phosphorylating substrates with proline in the P + 1 position.

In total, the CMGC family comprises 62 members (uniprot.org (accessed on 29 May 2023) [9]), which can be further subdivided into eight subfamilies [5] (Figure 1). CDKs and MAPKs constitute the two largest and most extensively studied families, with 21 and 14 members, respectively [10]. Other families include DYRKs (dual-specificity tyrosine (Y)-phosphorylation-regulated kinases; 10), CLKs (4), RCKs (tyrosine kinase gene v-ros cross-hybridizing kinase; 3), CDKL (cyclin-dependent kinase-like; 5), GSK (2), and SRPK (SR-specific protein kinase; 3). Most CMGC kinases are associated with specific biological functions, such as controlling the cell cycle (CDKs), determining cell fate (MAPKs), regulating multiple signaling pathways (GSKs), and facilitating RNA splicing (CLKs). The individual subfamilies and their unique features will be described in detail below.

Given the crucial roles of CMGC kinases in regulating various cellular processes, their dysregulation can contribute to cancer development and progression. In this review, we will delve into the importance of CMGC kinases in cancer biology, their involvement in cellular signaling pathways, protein–protein interactions, and the current state of kinase inhibitors targeting these kinases. Furthermore, we will discuss the challenges and future perspectives in targeting CMGC kinases for cancer therapy, including potential strategies to overcome resistance, the development of more selective inhibitors, and novel therapeutic approaches.

CMGC kinases have gained considerable attention in the context of targeted cancer therapies, with several FDA-approved inhibitors already available and numerous others in various stages of clinical trials. While these inhibitors have demonstrated efficacy and safety to varying degrees, challenges persist, such as resistance development and off-target effects. The need for selective kinase inhibitors and novel therapeutic approaches, including combination therapies, underscores the importance of continued research in this area. Furthermore, the discovery and validation of specific biomarkers for patient stratification and personalized medicine approaches are crucial for selecting the most appropriate therapeutic strategies for individual patients, ultimately improving treatment outcomes and minimizing potential side effects.

In summary, this review highlights the importance of CMGC kinases in eukaryotic cells and their roles in regulating critical cellular processes. As such, understanding the dysregulation of these kinases and their contribution to cancer development provides a foundation for the development of targeted therapies. By delving into the structure, function, and regulation of CMGC kinases, this review aims to provide a comprehensive overview of their involvement in cancer biology and therapeutic potential. The subsequent sections will explore each subfamily in greater depth and discuss the current state of research on kinase inhibitors targeting CMGC kinases, as well as the challenges and future directions in this area of cancer therapy.

## 2. CMGC Kinase Subfamilies

### 2.1. Cyclin-Dependent Kinases (CDKs)

CDKs or cyclin-dependent kinases form a family of 21 constitutively expressed serine/threonine kinases that regulate the cell cycle, transcription, mRNA processing, and the differentiation of cells [11]. As their name suggests, CDKs depend on cyclins, proteins whose levels constantly fluctuate in a tightly coordinated manner during the cell cycle. In CDK/cyclin complexes, CDKs function as catalytic subunits, while cyclins serve as regulatory subunits required for CDKs to exhibit enzymatic activity. The primary mechanism for regulating CDK enzymatic activity is the production and degradation rate of cyclins. The CDK activation loop contains residues for both cyclin and subsequent ATP binding. Like most eukaryotic protein kinases, CDKs are also regulated by the phosphorylation of the activation segment. Phosphorylation can either inhibit by interfering with ATP binding at the catalytic cleft or activate when the phosphorylation site serves as a substrate for CDK-activating kinases, including other CDKs. Activating phosphorylation enhances substrate binding by promoting full kinase activity [12,13].

CDK activity is downregulated by two families of small proteins that function as CDK inhibitors during the cell cycle. The INK4 protein family acts during the G1 phase, specifically inhibiting CDK4/CDK6 binding with cyclin D [11,14]. The broader-spectrum inhibitors of the Cip/Kip family inhibit CDK/cyclin complex activity throughout the cell cycle [11,14]. Based on evolutionary clustering, CDKs form kinase subfamilies that regulate separate cellular functions. CDKs 1–4 and 6 primarily regulate cell cycle progression, whereas CDKs 7–9, 11–13, and 19 have established roles in transcription [11]. Other CDKs have varying roles that will be discussed later.

#### 2.1.1. Cell Cycle Regulation by CDKs (CDK 1–4, and 6)

Most living organisms are unicellular, but others, like humans, consist of millions of cells. During the development of these multicellular organisms, multiple rounds of cell growth and division must occur, a process that continues throughout an individual’s life. This elegant, tightly regulated universal process of cell growth and division, where a cell duplicates its contents and divides into two, is known as the cell cycle. Under normal conditions, the cell cycle is tightly regulated at each stage through the activation or deactivation of various proteins. The mammalian cell cycle has five phases: G0 (resting state), G1 and G2 (RNA and protein synthesis), S (DNA replication), and M (mitosis and the completion of cell division). Multiple checkpoints ensure normal progression from one phase to another, with specific cyclin/CDK complexes controlling cell cycle progression. Briefly, the cyclin C/CDK3 complex assists cells in the G0–G1 transition ([15]; uniprot.org (accessed on 29 May 2023)). The cyclin D/CDK4 (CDK6) complex initiates G1 progression through the phosphorylation of the retinoblastoma protein, leading to the transcription of genes needed for DNA synthesis and further cell cycle progression (Figure 2A). The cyclin E/CDK2 complex serves as a gatekeeper in the G1/S transition and is responsible for the hyper-phosphorylation of the retinoblastoma protein. During the S phase, cyclin A accumulates, and the activity of the cyclin A/CDK2 complex is essential for S phase termination via phosphorylation of transcription factor E2F1. Cyclin A/CDK1 and cyclin B/CDK1 complexes are crucial for progression through the G2 and M phases [11,14]. CDK7 can also influence the cell cycle by acting as a CDK-activating kinase, phosphorylating, for example, CDK1 and CDK2 [16].

To study the functional roles of individual cell-cycle-related CDKs in vivo, multiple knockout mice have been generated. The analyses of knockout animals lacking one or more of the cell-cycle-related CDKs—CDK1 [17], CDK2 [18,19], CDK4 [20,21], and CDK6 [22]—have shown that only CDK1 is essential for embryonic cell division, leading to early embryonic lethality. CDK2, CDK4, and CDK6 are mainly needed for the development of distinct cell types, and the knockout animals develop normally into adulthood [23]. CDK4/CDK6 double-knockout mutant embryos die in utero [22].

#### 2.1.2. Transcription Initiation, Elongation, and Termination Regulation by CDKs (‘Transcriptional CDKs’: CDK 7–9, 11–13, and 19)

Transcription, the tightly regulated process of copying genetic information from DNA to RNA, relies on RNA polymerase II (RNA pol II) for all protein-coding genes. RNA pol II has a C-terminal domain (CTD) composed of polypeptide repeats, which are phosphorylated in a gene-specific manner to regulate its activity. The first step in transcription involves recruiting RNA pol II to the promoter of the transcribed gene, a task performed by the transcription preinitiation complex (PIC) that comprises several general transcription factors (TFIIA, TFIIB, TFIID, TFIIE, IFIIF, TFIIH), the Mediator complex, and RNA pol II itself (Figure 2B). The TFIIH complex, which includes CDK7, cyclin H, and MAT1, is crucial at the transcription start site. It unwinds the DNA, positions it correctly in the RNA pol II active site, and phosphorylates a serine (primarily Ser-5, but it can also be Ser-7) at the CTD of RNA pol II to initiate RNA production. As the transcript reaches a certain length, RNA pol II sheds most of the initial transcription factors, and new factors are recruited. During elongation, one of these newly recruited factors is the CDK9/cyclin T-containing positive transcription elongation factor, pTEFb. The CDK9 subunit phosphorylates another serine (Ser-2) in the CTD of RNA pol II. CDK12 interacts with cyclin K and has been shown to phosphorylate the same serine-2 in the CTD. CDK12, CDK11, and CDK13 also play roles in alternative RNA splicing (uniprot.org (accessed on 29 May 2023); [24,25]). The last step of transcription is termination, which releases both RNA pol II from the template DNA and mRNA from the transcriptional complex. The presence of CDK8 and the CDK19-containing Mediator complex is also necessary for RNA pol II phosphorylation, although the primary function of the Mediator is to transfer gene-specific regulatory signals from multiple transcription factors to RNA pol II [26]. To date, knockout animal models of transcriptional CDKs are available only for CDK7 [27], CDK8 [28], CDK11 [29], and CDK12 [30]. All of these knockouts are embryonic-lethal during early development.

#### 2.1.3. Other CDKs (CDK5–10, 14–18, and 20)

CDK5 is expressed in post-mitotic cells like neurons [31]. The presence of CDK5 is essential for the developing brain [32], and later, in the adult brain, it is involved in multiple neuronal processes, including learning and memory, survival, synaptic plasticity, and pain signaling [33]. In the brain, CDK5 is activated by interaction with the non-cyclins CDK5R1 (p35) or CDK5R2 (p39). Cyclins D1, D3, and E have been reported to form a complex with CDK5 [33], and in 2009 [34], cyclin I was identified as a new CDK5 activator. Recently, cyclin I-like (CCNI2), a homolog of cyclin I, was shown to activate CDK5, affect its cellular localization, and has a role in cell cycle progression and cell proliferation [35].

The functional role and activating cyclin of CDK10 were only recently detected [36]. Previously, the silencing of CDK10 was linked to ETS2-driven increased expression of c-RAF, leading to MAPK pathway activation, and a loss of estrogen responsiveness of breast cancer cells, causing tamoxifen resistance in certain breast tumors [37]. The study of Guen (2013) showed that cyclin M interacts with CDK10, and this complex phosphorylates the ETS2 transcription factor, positively controlling ETS2 degradation by the proteasome.

CDK14 to CDK18 belong to the PFTAIRE and PCTAIRE kinase subfamilies (HUGO Gene Nomenclature Committee, genenames.org, accessed on 29 May 2023). CDK14 and CDK16 are activated at the plasma membrane by cyclin Y and participate in a multitude of signaling cascades, such as the Wnt pathway [38,39]. This pathway is known to control transcription, and its regulators are also important during mitosis. Notably, cyclin Y expression has been shown to peak during the G2-M phase of the cell cycle, suggesting these enzymes play a role in cell division [11]. CDK15, CDK17, and CDK18 are the least studied CDKs. CDK15 has been shown to phosphorylate survivin and lead to a reduction in tumor-necrosis-factor-related apoptosis-inducing ligand (TRAIL)-induced apoptosis [40]. The elevated phosphorylation of CDK17 and CDK18 was recently identified in a neuroblastoma cell line expressing amyloid precursor protein (APP) [41]. Cyclin-dependent kinase 20 (CDK20) plays a key role in the regulation of primary cilia, essential sensory organelles in cells. It controls ciliary length and contributes to cilia disassembly. The overexpression of CDK20 results in shorter cilia, whereas its knockdown leads to longer cilia and more ciliated cells, due to increased cilia stability in the absence of kinase activity [42]. Given the critical role of cilia in various cellular processes and diseases, understanding CDK20′s function provides important insights into these areas.

In sum, the cyclin-dependent kinases (CDKs) are a diverse family of proteins that play crucial roles in regulating cell cycle progression, transcription, and various signaling pathways. The functions of CDKs are tightly regulated through interactions with their activating cyclin partners and various phosphorylation events. Given their significant roles in cellular processes, the dysregulation of CDKs has been implicated in numerous diseases, including cancer and developmental abnormalities. Consequently, CDKs have emerged as potential therapeutic targets, and efforts are being made to develop specific inhibitors that can modulate their activity. In 2020, the FDA approved the CDK4/6 inhibitor abemaciclib in combination with endocrine therapy for the adjuvant treatment of hormone-receptor-positive, HER2-negative, high-risk early breast cancer [43].

Studies have revealed a critical role for CDK2 in renal carcinogenesis. The overexpression of CDK2 has been found in clear cell renal cell carcinoma (ccRCC), the most common type of kidney cancer, and it is associated with a higher grade and stage of the disease [44]. The study showed that higher CDK2 expression promotes cell proliferation and inhibits apoptosis in ccRCC. Furthermore, CDK2 overexpression has been associated with poor prognosis in patients with ccRCC, indicating its potential as a prognostic biomarker. The role of CDK1 and -2 activity as a biomarker in renal cell carcinoma has been studied [45] and downregulating CDK2 by miRNA in a mouse model suppressed tumor growth [44].

Future research will likely continue to uncover novel functions and regulatory mechanisms involving CDKs, providing new insights into their roles in both physiological and pathological processes. A deeper understanding of CDKs will aid in the development of more effective therapeutic strategies to target these kinases and combat cancer.

### 2.2. Mitogen-Activated Protein Kinases (MAPKs)

The mitogen-activated protein kinase (MAPK) family consists of 14 serine/threonine kinases that can be divided into atypical and conventional kinases.

The atypical mitogen-activated protein kinases (MAPKs) form a distinct, unique subgroup within the larger MAPK family, a major regulator of cellular processes. This group includes extracellular signal-regulated kinase 3 (ERK3), ERK4, ERK7, and Nemo-like kinase (NLK) [46]. Unlike their typical counterparts, atypical MAPKs feature a divergence from the canonical MAPK architecture in their kinase domain, deviating from the conserved TXY motif crucial for activation [47]. Moreover, they exhibit unique activation and regulatory mechanisms, such as autophosphorylation [48]. Their specific physiological roles and functions remain more undefined compared to classical MAPKs. However, emerging evidence links atypical MAPKs to several critical cellular processes, including cell differentiation, proliferation, motility, and apoptosis [49]. For instance, ERK3 and ERK4 are associated with cell migration and actin cytoskeleton reorganization, while ERK7/8 plays roles in cell cycle progression and response to stress [50]. In spite of this, the mechanisms by which atypical MAPKs exert their influence and their potential implications in human diseases, such as cancer, are areas of ongoing research and discovery.

The conventional kinases can be further subdivided into four subfamilies: (1) extracellular signal-regulated kinases 1/2 (ERK1/2), (2) c-Jun amino-terminal kinases 1, 2, and 3 (JNK1-3), (3) p38 MAPKs, and (4) extracellular signal-regulated kinase 5 (ERK5). These enzymes are characterized by a system in which three kinases act sequentially to phosphorylate the downstream kinase (Figure 2C). Activators for ERKs are MEK1 and MEK2; for JNKs-, MKK4 and MKK7; for p38-, MKK3 and MKK6; and for ERK5-, MEK5. MAPKs are highly conserved, respond to extracellular signaling cues, and regulate numerous essential cellular processes such as proliferation, differentiation, stress responses, and the transcriptional control of CDKs in organisms ranging from yeast to humans. Over the past few decades, extensive research has been conducted on MAPKs, from their substrates and functions to their roles in health and cancer.

Extracellular signal-regulated kinases 1/2 respond to various stimuli, including growth factors, insulin, cytokines, and carcinogens [50,51]. They are components of the cell surface receptor (receptor tyrosine kinase)/RasGTP/Raf/MEK/ERK signaling cascade. Components of this signaling cascade are essential for proper cell proliferation. ERKs function both in the cytoplasm to promote cell proliferation (G1 to S phase transition) and in the nucleus to phosphorylate and activate numerous transcription factors (TFs) [50,52]. Upon TF activation, c-Jun is expressed and stabilized by direct phosphorylation by ERK1/2 [53], allowing association with c-Fos and the subsequent formation of the AP-1 complex. AP-1 is essential for cyclin D1 expression, which interacts with CDKs to drive G1 to S transition.

c-Jun N-terminal kinases are highly homologous (>85%) but have a distinct tissue distribution. JNK1 and 2 are ubiquitously expressed, while JNK3 expression is mainly limited to the brain. In contrast to ERKs, JNKs are primarily activated by stress signals such as oxidative stress, radiation, and DNA-damaging agents. JNKs are mainly localized in the cytoplasm, but their identified substrates are mostly TFs, including c-Jun, p53, STAT3, and c-Myc. The phosphorylation of c-Jun leads to AP-1 complex formation and thus the transcription of cyclin D1, promoting cell cycle progression, similar to ERK1/2 [54]. To date, only a few cytoplasmic interaction partners of JNKs have been identified [50,55].The p38 subfamily consists of four members (α, β, γ, δ), which respond to various environmental stress stimuli and cytokines such as interleukin-1 and tumor necrosis factor α(TNF). Interestingly, p38 both regulates the production of cytokines and responds to them. Other targets of p38 regulation are TFs and other protein kinases. Based on observations of p38 activation, it plays a role in inflammation, cell cycle regulation, and apoptosis [50,56].ERK5 (BMK1 or big MAP kinase 1) has a kinase domain similar to ERK1/2, sharing 51% similarity with ERK2. ERK5 is essential during normal embryogenesis [57]. An upstream activator of ERK5 is MEK5, whose expression is elevated in metastatic prostate cancer [58]. Similar to ERK1/2 and JNK, ERK5 also promotes cyclin D1 expression and cell cycle progression [59], as well as plays a crucial role in the maintenance of mitochondrial function and neuronal survival [60]. ERK5 is involved in various cellular processes, including cell survival, differentiation, and angiogenesis. Its activation has been linked to growth factors, oxidative stress, and other extracellular stimuli.The MAPK pathway has a critical role in cancer biology, extending beyond the extensively studied BRAF mutation in melanoma. The MAPK/ERK pathway, for instance, has been implicated in colorectal cancer, with mutations in KRAS and NRAS genes leading to its persistent activation, promoting uncontrolled cell proliferation and tumor growth [61]. These mutations, unfortunately, render the tumors resistant to EGFR-targeted therapies, highlighting the need for novel therapeutic strategies [62].Additionally, the JNK MAPK pathway, associated primarily with responses to stress signals and apoptosis, has shown links to cancer biology. Aberrations in JNK signaling can lead to an imbalance between cell proliferation and death, thereby contributing to oncogenesis. For instance, overactive JNK signaling has been found in several cancers, including breast and gastric cancer, often correlating with a worse prognosis [63]. Furthermore, the p38 MAPK pathway, typically associated with inflammation and cell differentiation, is also relevant in cancer research. Its complex, dual role in tumorigenesis is being unraveled; while its activation can suppress tumor growth by promoting cell cycle arrest and apoptosis, chronic activation can also enhance cancer cell survival, contributing to chemoresistance [64].

In summary, the MAPK family comprises a diverse group of kinases that play crucial roles in cellular signaling pathways. These kinases are involved in various cellular processes, including proliferation, differentiation, and stress responses, and are integral to the transcriptional control of CDKs. Alterations in the function of the components of the cellular signaling cascade can lead to cancer. Thus, understanding the mechanisms and functions of MAPKs is essential for advancing our knowledge of cellular processes and uncovering potential therapeutic targets in various diseases.

### 2.3. Glycogen Synthase Kinase-3 (GSK-3)

Glycogen synthase kinase-3 is a conserved, ubiquitously expressed serine/threonine kinase with unique characteristics, such as its high activity in resting cells. Humans have two forms of GSK-3s: GSK-3α and GSK-3β, which share an 85% overall sequence homology [65]. The kinase domain of these kinases is highly homologous (98%), while the N- and C-terminal domains differ [65]. GSK-3α and GSK-3β have distinct functions, as evidenced by the fact that homozygous GSK-3β knockout mice are embryonic-lethal, whereas GSK3α knockout mice are viable [66,67].

Unlike the CDK and MAPK families, GSK activity is regulated through multiple levels and mechanisms, ranging from phosphorylation by other kinases, autophosphorylation, and priming phosphorylation of the target substrate by another kinase [68,69,70]. In most cases, the phosphorylation of serine residues inhibits GSK’s kinase activity, while tyrosine phosphorylation increases its activity [71]. GSK-3 (mainly GSK-3α) has been extensively studied and is known to function in various signaling pathways, including Wnt, Notch, and Hedgehog (proliferation), as well as growth factors affecting differentiation/survival. It has been implicated in multiple roles in many human pathological conditions, ranging from neurodegenerative diseases to diabetes and several types of cancers [70,72].

GSK is a critical component of the canonical Wnt signaling pathway, which is essential during normal development and is often dysregulated in cancers. The main events during Wnt-β-catenin signaling involve a central component known as the destruction complex that regulates cytoplasmic β-catenin levels. This complex comprises tumor suppressors Axin and adenomatosis polyposis coli (APC), as well as casein kinase-1α (CK1α) and GSK-3 [73]. In the absence of the Wnt-ligand, β-catenin is assembled into this complex and phosphorylated by CK1 and GSK-3, marking it for proteasomal degradation and thus preventing the transcription of β-catenin-dependent genes. When Wnt binds to its receptor Frizzled, the destruction complex cannot phosphorylate and ubiquitinate β-catenin, allowing it to translocate to the nucleus, where it forms complexes with transcription factors and promotes transcription. Phosphorylation by GSK-3 is also essential for the targeting of multiple other proteins for proteasomal degradation [72,74]. In addition to its role in Wnt signaling, GSK-3 has been implicated in other critical signaling pathways. For example, GSK-3 plays a role in the insulin signaling pathway, which is essential for glucose homeostasis. Abnormalities in GSK-3 function have been linked to insulin resistance and the development of type 2 diabetes [75]. GSK-3 inhibitors have shown promise in preclinical studies as potential therapeutic agents for the treatment of type 2 diabetes [76].

GSK-3 has been found to be overactive in various types of cancers, including colon, breast, and prostate cancer. The overactivation of GSK-3 can lead to increased cell proliferation, migration, and invasion, as well as reduced apoptosis [70]. Targeting GSK-3 may represent a potential therapeutic strategy for cancer treatment. Preclinical studies have shown that GSK-3 inhibitors can suppress tumor growth and metastasis, as well as sensitize cancer cells to chemotherapy and radiation therapy [72].

In summary, GSK-3 is a multifunctional kinase that plays a pivotal role in numerous signaling pathways and cellular processes. The dysregulation of GSK-3 activity has been implicated in various pathological conditions. Consequently, GSK-3 has emerged as a potential therapeutic target, and the development of GSK-3 inhibitors may hold promise for the treatment of these diseases. However, due to the pleiotropic nature of GSK-3, further research is needed to understand the precise mechanisms of GSK-3 regulation and its involvement in different pathologies to develop safe and effective therapeutic strategies.

### 2.4. Dual-Specificity Tyrosine (Y)-Phosphorylation-Regulated Kinases (DYRKs)

DYRK is a relatively large, mammalian, dual-specificity protein kinase family that, based on homology analysis, consists of three subfamilies with a total of 10 members: DYRK (DYRK1A-B, 2–4), HIPKs (homeodomain-interacting protein kinase 1–4), and PRP4s (pre-mRNA processing protein 4 kinase).

#### 2.4.1. DYRK1–4

The DYRK subfamily members are autophosphorylated on tyrosine within the activation loop during their translation, resulting in a constitutively active mature protein [77]. Autophosphorylation of this conserved tyrosine is essential for achieving full kinase activity [78]. As DYRKs are not subject to classical MAPK-like regulation by an upstream protein kinase, other regulatory mechanisms have been proposed. All of these subfamily members experience changes in their cytoplasmic versus nuclear localization, which are speculated to limit substrate accessibility and, thus, regulate DYRK1–4 [79]. Other regulatory mechanisms for DYRKs might include alterations in gene expression and protein abundance or interactions with regulatory proteins. Thus far, DYRKs have not been shown to undergo regulation by reversible phosphorylation or dephosphorylation [79].

The major function of mammalian DYRKs is the regulation of the cell cycle. Both DYRK1A and DYRK1B can be classified as survival- and differentiation-promoting factors. DYRK1A has been shown to control the length of the G1 phase by regulating cyclin D1 levels and promoting the transition between quiescence and differentiation [80]. DYRK1B is active in quiescent cells, preventing entry into G1 by stabilizing CDK inhibitors and destabilizing cyclin D [81,82]. In apoptosis, DYRK1A, DYRK3, and DYRK2 have opposite roles, as DYRK1A and DYRK3 phosphorylate either pro-apoptotic proteins or their inhibitors, leading to the inhibition of apoptotic activity [83,84], while the phosphorylation of p53 by DYRK2 (or HIPK2) promotes apoptosis [85].

Another common characteristic of the DYRKs is their action as priming serine/threonine kinases for subsequent phosphorylation by GSK-3, thus targeting proteins for proteasomal degradation [79]. To date, DYRK family members have been identified as priming kinases for oncoproteins c-Jun and c-Myc, in addition to transcription factor GLI2 [9,85]. Although the functions of the fourth member of this subfamily, DYRK4, are ambiguous, based on the molecular roles of other DYRKs, one can conclude that the members of this subfamily are involved in cell survival, cell differentiation, gene transcription, and translation [79].

#### 2.4.2. Homeodomain-Interacting Protein Kinase (HIPK)

HIPKs, like DYRKs, are autophosphorylated to reach full kinase activity [86]. HIPK1–3 are highly homologous, whereas the later identified fourth member, HIPK4, is only related to others in its catalytic domain [87]. Most of the knowledge about HIPKs comes from HIPK2, which is known to act in the cell cycle, apoptosis, and responses to DNA damage, mostly via interactions with transcription factors. The activity of HIPK2 itself is orchestrated by multiple post-translational modifications and caspase-mediated cleavage [86]. Due to structural similarity, HIPK1 is thought to have a similar role in cells to HIPK2 [88]. A recent study identified HIPK2 as a facilitator of Disheveled phosphorylation, maintaining cells responsive to Wnt ligand stimulation [89]. In 2019, a study revealed the role of HIPK1 in the regulation of cellular stress response and its potential as a therapeutic target for the treatment of glioblastoma [90]. The molecular functions of both HIPK3 and, especially, HIPK4 are still rather uncharacterized.

#### 2.4.3. Pre-mRNA Processing Protein 4 Kinase (PRP4)

PRP4 is a subfamily of ubiquitously expressed kinases that regulate pre-mRNA splicing and act as spindle assembly checkpoint proteins [91,92]. The kinase domain of PRP4 was identified as essential for pancreatic and colorectal tumor cell viability [93]. The PRP4 family members play a critical role in maintaining the proper function of spliceosomes, the cellular machinery responsible for splicing introns from pre-mRNA [94]. PRP4 kinase phosphorylates several spliceosome components, including the U4/U6-U5 tri-snRNP, to facilitate the assembly and disassembly of the spliceosome [95]. A 2016 study reported the role of PRP4 in regulating the pre-mRNA splicing of the POLD1 gene, which is involved in DNA replication and repair, suggesting a potential link between PRP4 and cancer development [96]. Due to the association between aberrant splicing and various diseases, PRP4 kinases are interesting targets for further investigation.

### 2.5. Cdc2-like Kinase (CLK) and Other Less-Studied Kinases

The Cdc2-like kinase (CLK) family consists of four dual-specificity kinases (CLK1–4) that autophosphorylate on tyrosine residues and specifically phosphorylate serine/threonine residues on their substrates [97]. CLKs are found in both the cytoplasm and the nucleus, where they phosphorylate the arginine- and serine-rich (RS) domains of serine/arginine (SR) splicing factor proteins, thereby controlling their nuclear distribution [98]. SR splicing factor proteins are extensively phosphorylated by multiple kinases, and the proper phosphorylation status has been shown to be essential for SR protein activity [99]. These proteins play a crucial role in alternative mRNA splicing, a fundamental process responsible for generating the complex human proteome during normal development.

In a recent study, CLKs were identified as upstream regulators of Aurora B, acting as novel components in the final steps of cytokinesis, the division of the cytoplasm into two daughter cells [100]. However, as Petsalaki and Zachos concluded, our current understanding of CLKs’ cellular functions remains limited [100]. A 2017 study reported that CLK inhibition could modulate the alternative splicing of Mcl-1, an anti-apoptotic protein, and promote cancer cell death, suggesting a potential therapeutic approach for cancer treatment [101]. Given the importance of alternative mRNA splicing in the regulation of gene expression and the emerging roles of CLKs in cell division, further investigation into the functions and regulation of CLKs could provide valuable insights into their role in various cellular processes and their potential as therapeutic targets in human diseases.

### 2.6. SR-Specific Protein Kinase (SRPK)

Serine–arginine protein kinases (SRPKs) specifically phosphorylate the serine residues in serine/arginine (SR) dipeptides. They are primarily localized in the cytoplasm, where these constitutively active kinases regiospecifically phosphorylate SR splicing factors on multiple serines, inducing the translocation of these factors into the nucleus. In the nucleus, SR splicing factors are further phosphorylated by CLKs [102,103]. The change in the cellular localization of SR proteins is essential for phosphorylation by CLKs [98]. As mentioned earlier, SR protein phosphorylation is necessary for correct mRNA splicing and maturation.

Over 100 SR-domain-containing proteins have been identified in the human genome [104], indicating that there are many open questions regarding the functional roles of SRPKs in mammalian cells. SRPK1 expression is elevated in breast, colon, and pancreatic cancers [105], as well as in acute T-cell leukemia [106]. A 2020 study identified SRPK1 as a potential therapeutic target in glioblastoma, with the SRPK1 inhibitor SPHINX31 showing promising results in preclinical models [107]. SRPK2 has a specific role in phosphorylating apoptosis-promoting protein ACIN1 and increasing cyclin A1 expression in leukemia cells [108]. Given the importance of SRPKs in the regulation of alternative splicing and their association with various types of cancer, further studies on the functional roles of these kinases and their potential as therapeutic targets in human cancer are warranted.

### 2.7. Tyrosine Kinase Gene v-Ros Cross-Hybridizing Kinase (RCK)

The RCK serine/threonine-protein kinase family consists of three kinases: MAK, ICK, and MOK. The functions of these kinases are poorly understood. Structurally, the RCK family resembles both MAPKs and CDKs [5]. MAK and ICK autophosphorylate on a tyrosine residue, but they need a second phosphorylation on their MAPK-like motif in their activation loop by an upstream kinase for full enzymatic activity [109,110,111]. Different from MAPK, the activating kinase for MAK and ICK is cell-cycle-related kinase (CCRK), but for MOK, it remains unknown [111,112].

MAK (male germ cell-associated kinase) is mainly expressed in testicular germ cells during spermatogenesis and in the retina. In the retina, MAK localizes to connecting cilia in photoreceptor cells, negatively regulates the length of their cilia, and is essential for the survival of these cells [113]. Not surprisingly, MAK mutations are associated with retinitis pigmentosa, a photoreceptor degeneration disease in the retina [114,115].ICK (intestinal cell kinase) is highly conserved, constitutively, and widely expressed. Similarly to MAK, it negatively regulates ciliary length and is identified as an essential component of sonic hedgehog signaling [116]. These two factors seem to be the underlying cause of human ECO syndrome, a multi-organ illness affecting the endocrine, cerebral, and skeletal systems, caused by a missense mutation in the ICK gene [117]. In 2017, a study reported the role of ICK in colorectal cancer progression and its potential as a therapeutic target for the treatment of colorectal cancer [118].

The cellular function of the third RCK, MOK (MAPK/MAK/MRK-overlapping kinase), is largely unknown. To date, only the expression of MOK in mouse wild-type and cancerous intestinal cells has been analyzed by Western blotting, showing the downregulation of MOK in adenomas [80]. This observation suggests that MOK might have a role in intestinal cancer development, but further studies are needed.

### 2.8. Cyclin-Dependent Kinase-like (CDKL)

This family of five members of serine/threonine kinases with homology to MAPK and CDK families is relatively uncharacterized. Studies have focused on CDKL5, a ubiquitous protein mainly expressed in the brain, testes, and thymus [119]. In the context of cancer, CDKL proteins are now emerging as potential players. Studies suggest that the dysregulation of CDKL5 may contribute to tumorigenesis in certain cancers. For instance, the aberrant expression of CDKL5 has been suggested in glioma and some evidence exists on its role in breast cancer, indicating its possible role in disease progression and patient prognosis [120].

Similarly, another family member, CDKL1, has been associated with colorectal cancer, with its overexpression correlating with poor survival outcomes [121]. These preliminary findings suggest that CDKL proteins could potentially act as novel markers for cancer diagnosis or targets for therapeutic intervention.

## 3. Protein Kinase Therapeutics—Kinase Inhibitors

The protein kinase domain is among the most commonly encountered domain in known cancer genes [122]. As described earlier, most of the CMGC kinases affect various cellular signaling pathways that control critical processes like cell cycle progression, proliferation, differentiation, apoptosis, or survival. Often, abnormal phosphorylation is either the cause or consequence in cancer [123]. Therefore, protein kinases are among the most studied druggable targets in pharmacological research [124,125]. In human cancers, kinases are often found overexpressed or overactive, due to a vast array of genetic and epigenetic events, including point mutations, chromosomal gene amplifications (copy number alterations), and chromosomal translocations giving rise to gene fusions [126] (Table 1).

### 3.1. Oncogenic Relations of CMGC Family Members

Combined data from six recent studies identified a total of 1100 genes that drive cancer development. From this list, an enrichment of protein kinases was observed (91 kinases were present) [149]. Six of them (6/91; ~7%) belonged to the CMGC kinase family [149], involving family members of CDK, MAPK, and DYRK, namely CDK4, CDK6, CDK12, MAPK1 (Erk2), MAPK8 (JNK1)), and DYRK1A. Expectedly, the alterations observed in these genes were either point mutations or copy number changes [149]. Other studies have also identified amplified CDK12 expression in breast cancers and inactivated in ovarian cancers [150,151,152]. In addition, over-expression of CDK7 together with MAT1 and cyclin H was detected in >900 estrogen-receptor-positive breast cancer samples [153].

The deregulation of MAPK signaling pathways can result in the activation of oncogenic signaling, leading to increased cell proliferation, survival, and migration [154]. Mutations in components of the MAPK pathways, such as RAS and RAF, can lead to the constitutive activation of MAPK signaling, promoting cancer development and progression [155]. Aberrant activation of MAPK signaling has been implicated in various cancer types, including melanoma, colorectal, and lung cancers [154].

DYRK1B overexpression has been observed in pancreatic and ovarian cancer [156], and in osteosarcomas and rhabdomyosarcomas [141,142]. Interestingly, the depletion of DYRK1B drove pancreatic and ovarian cancer cells to apoptosis [157,158].

Dysregulation of GSK-3 can contribute to cancer development by affecting various cellular processes, such as cell cycle regulation, differentiation, and apoptosis [69]. GSK-3 can act as a tumor suppressor or an oncogene, depending on the cellular context and specific signaling pathways involved [159]. For example, GSK-3 promotes the degradation of β-catenin, a key mediator of the Wnt signaling pathway, thereby preventing the activation of oncogenic Wnt target genes [69]. However, GSK-3 can also promote cancer progression by stabilizing the oncoprotein c-Myc and enhancing its activity [69].

The aberrant activation of CLKs has been associated with cancer-associated splicing alterations, which can lead to the production of aberrant protein isoforms that contribute to tumor progression [160]. For example, CLKs have been implicated in the regulation of alternative splicing events that promote cell survival, angiogenesis, and metastasis in various cancer types [161,162]. Moreover, the overexpression of CLKs has been observed in several cancer types, such as breast and prostate cancers, and is associated with poor prognosis [160].

### 3.2. Therapeutic Targeting of CMGC Kinases

Based on the common structural features of protein kinases, most protein kinase inhibitors target the ATP-binding site in the activation segment, thus competing with ATP. The second-generation inhibitors target protein kinase in a specific conformation and are thought to be more specific. However, there are highly selective and broad range inhibitors within both generations of kinase inhibitors [163]. Unfortunately, a common shortcoming of kinase inhibitors is the relatively low percentage of patients presenting a satisfactory response [125]. The best clinical outcome is usually reached with a polypharmacological approach, using drug combinations either within a single kinase pathway or targeting parallel kinase pathways in preselected patient populations [125,126] (Table 2).

#### 3.2.1. CDK Inhibitors

Several CDK inhibitors have been developed to target specific CDKs involved in cell cycle regulation and other cellular processes [164]. The first US Food and Drug Administration (FDA)-approved CMGC family kinase inhibitor was palbociclib, a selective CDK4/6 inhibitor [165]. Altogether, the FDA has approved three CDK4/6 inhibitors, palbociclib, ribociclib, and abemaciclib, for the treatment of hormone-receptor-positive, HER2-negative, advanced or metastatic breast cancer [165,166,167]. These inhibitors have shown promising results in clinical trials, improving progression-free survival and overall response rates. Further research is exploring their potential in other cancer types and combination therapies.

#### 3.2.2. MAPK Inhibitors

The success of kinase inhibitors targeting the MAPK pathway components, such as BRAF and MEK inhibitors, has demonstrated the therapeutic potential of targeting MAPK signaling in cancer [132,133]. The FDA has approved BRAF inhibitors (vemurafenib, dabrafenib) and MEK inhibitors (trametinib, cobimetinib) for the treatment of metastatic melanoma harboring BRAF mutations. Additionally, combined BRAF and MEK inhibition has shown improved outcomes compared to single-agent therapy [133]. Efforts to develop MAPK inhibitors targeting other components of the MAPK pathways are ongoing.

#### 3.2.3. DYRK Inhibitors

A polyphenolic green tea flavonol, epigallocatechin-3-gallate (EGCG), is a DYRK1A inhibitor that was shown to normalize DYRK1A activity in Down syndrome patients in a pilot study reversing cognitive deficits [168]. The inhibition of DYRK1A is an enchanting therapeutic approach for the treatment of cognitive impairment in Down syndrome together with inhibiting tau hyperphosphorylation in Alzheimer’s disease [124]. DYRK-family inhibitors have also been shown effective in vitro in reducing proliferation and inducing apoptosis either as single agents or as dual-inhibitors targeting CKL1 [124].

#### 3.2.4. GSK Inhibitors

A plethora of GSK-3 inhibitors have been published and studied in clinical trials for the treatment of cancer or Alzheimer’s disease [169]. However, the results of the studies have been controversial, and none of the inhibitors have made it into clinical use [169]. One of the studied inhibitors, tideglusib, showed promising preclinical results in different cancer types, such as glioblastoma and pancreatic cancer [69,170]. Further development and clinical evaluation of GSK-3 inhibitors are needed to determine their therapeutic potential in cancer treatment.

#### 3.2.5. CLK Inhibitors

Several small molecule inhibitors targeting CLKs are being investigated for their ability to modulate alternative splicing and their potential as anticancer agents [171]. One such CLK inhibitor, TG-003, has shown promising effects in preclinical studies, modulating alternative splicing and reducing tumor cell viability [172]. Still, no CLK inhibitors have been approved for clinical use.

All in all, the development of kinase inhibitors targeting CMGC kinases has shown promise in cancer treatment (Table 2). The approval of CDK4/6 and MAPK pathway inhibitors demonstrates the potential of targeting these kinases for therapeutic intervention. Ongoing research efforts aim to develop novel inhibitors targeting GSKs and CLKs and explore their therapeutic potential in cancer and other diseases. Future research should focus on overcoming resistance to kinase inhibitors, developing more selective inhibitors with fewer off-target effects, improving patient stratification, and exploring novel therapeutic strategies, such as targeting protein–protein interactions and exploiting synthetic lethality. Addressing these challenges will pave the way for more effective and personalized cancer treatments based on CMGC kinase inhibition.

### 3.3. Challenges in Targeting CMGC Kinases

#### 3.3.1. Resistance to Kinase Inhibitors

One major challenge in targeting CMGC kinases is the development of resistance to kinase inhibitors. Resistance can occur through various mechanisms, such as the acquisition of secondary mutations in the targeted kinase, the activation of compensatory signaling pathways, or alterations in drug metabolism [173,174]. Strategies to overcome resistance include the development of next-generation inhibitors with improved target selectivity, combination therapies targeting multiple signaling pathways, or sequential treatment strategies [175,176].

#### 3.3.2. Off-Target Effects and Toxicity

Kinase inhibitors may have off-target effects, leading to unwanted toxicity and limiting their therapeutic window. To address this issue, researchers are focusing on developing more selective inhibitors with higher specificity for their target kinases [177]. Additionally, understanding the molecular basis of these off-target effects can guide the development of inhibitors with fewer side effects.

### 3.4. Approaches to Overcome the Challenges

#### 3.4.1. Patient Stratification and Biomarkers

The identification of predictive biomarkers and proper patient stratification is essential for the successful implementation of targeted therapies. Efforts to identify biomarkers that predict response to CMGC kinase inhibitors can help select patients who are more likely to benefit from these treatments [178]. Additionally, the development of non-invasive techniques, such as liquid biopsies, can facilitate the real-time monitoring of treatment response and early detection of resistance [179].

#### 3.4.2. Exploiting Protein–Protein Interactions

Another approach to overcome the challenges associated with kinase inhibitor resistance is to target protein–protein interactions involving CMGC kinases. This strategy may provide a more selective way to modulate CMGC kinase activity and minimize off-target effects [180]. Further research is needed to identify and validate novel protein–protein interaction targets in CMGC kinase signaling pathways.

#### 3.4.3. Combination Therapies and Synthetic Lethality

Combination therapies that target multiple signaling pathways or exploit synthetic lethality can help overcome resistance and enhance the efficacy of CMGC kinase inhibitors [181] (https://clinicaltrials.gov/, accessed on 8 May 2023) (Table 3). Synthetic lethality arises when the simultaneous inhibition of two genes or pathways leads to cell death, while the inhibition of either one alone is insufficient [182]. Identifying synthetic lethal interactions involving CMGC kinases may reveal new therapeutic opportunities in cancer treatment.

As our understanding of the molecular mechanisms and protein–protein interactions involving CMGC kinases expands, so too will the opportunities for the development of more selective and effective therapeutic strategies.

## 4. Protein–Protein Interactions of the CMGC Kinases

CMGC kinases participate in various protein–protein interactions that are crucial for their activity, functions, and signaling pathways. Understanding these interactions is essential to elucidate the molecular mechanisms governing CMGC-kinase-mediated cellular processes. In this section, we will discuss some of the known interactions involving CMGC kinases and their importance (Figure 3).

CMGC kinases contribute to the development and progression of various diseases. Therefore, understanding the intricate network of protein–protein interactions involving CMGC kinases can provide valuable insights into the molecular mechanisms underlying their functions and offer potential therapeutic targets for the treatment of diseases associated with their dysregulation. For instance, the development of small molecule inhibitors that specifically target the protein–protein interactions of CMGC kinases could represent a promising approach to modulate their activity and restore normal cellular functions in disease settings [180].

The development of small molecule inhibitors that target the protein–protein interactions (PPIs) of these kinases has emerged as a promising strategy in cancer therapeutics. Traditional kinase inhibitors target the ATP-binding site of kinases. However, this site’s high conservation across different kinases challenges the development of inhibitors with high specificity. In contrast, the PPI interfaces are often less conserved, presenting an opportunity for creating more specific inhibitors [183,184]. Small molecule inhibitors act by obstructing the critical protein interactions necessary for the activation or functioning of these kinases. For instance, inhibitors may preclude a kinase from interacting with its activator or substrate, consequently inhibiting the kinase’s activity and downstream signaling [185]. In cancer, these inhibitors can restore normal cellular functions by blocking the aberrant signaling engendered by dysregulated kinases. For example, if a kinase is overactive in a cancer cell, promoting uncontrolled cell proliferation, an inhibitor that obstructs this kinase’s interactions could impede or slow down the proliferation [186]. Considerable progress has been made in the development of such inhibitors. High-throughput screening techniques have enabled researchers to identify small molecules that can bind to the PPI interfaces of CMGC kinases [185]. Furthermore, advances in structural biology have provided intricate details of these kinases’ 3D structures and their complexes with interacting proteins, aiding in the design of more effective inhibitors [187]. Nonetheless, challenges persist in developing PPI inhibitors. The interfaces involved in PPIs are often expansive and flat, complicating the ability of small molecules to bind with high affinity. Also, these inhibitors must navigate cellular barriers to reach their intracellular targets [186].

Most of the studies regarding kinase interactions have focused on a single kinase and its function or dysfunction in a disease. Additionally, the names of the CDK kinases were unified as late as in 2009 [10], and that might have contributed to the slowness of studies of some these kinases. It is, after all, rather difficult to assign a kinase to this interesting family when the name is very different, such as PFTK1 or CRKRS.

### 4.1. Affinity Purification and BioID Proximity Labeling

The first high-throughput screenings of protein–protein interactions were performed using the yeast two-hybrid technique [188]. This method allows the identification of binary interactions, thus assigning proteins to a biological context. Also, human protein–protein interactions were first studied in the yeast two-hybrid system as an important intermediate step towards a more systematic and comprehensive analysis of human interactome [189]. The next development in interactomics was the detailed characterization of yeast protein complexes utilizing affinity purification coupled to mass spectrometry (AP-MS) [190,191]. In 2009, Glatter and co-workers [192] presented a sensitive, reproducible, high-throughput AP-MS workflow for human cells. The general workflow of an AP-MS experiment has three steps: (1) expression of the epitope-tagged bait protein, (2) single- or double-step affinity purification of the protein complexes, and (3) analysis of the complex components with liquid chromatography coupled to a mass spectrometer.

This system offers several significant advantages. Some key benefits include the utilization of Gateway compatible human Orfeome [193] collections, the establishment of isogenic cell lines, the capability of inducible expression of bait proteins at nearly physiological levels, resulting in high yield, and reproducibility of affinity purification. Additionally, this system facilitates direct, “gel-free” analysis with LC-MS, enhancing its potential for precise and comprehensive proteomic studies [192]. Later, this approach was also shown to be robust and highly reproducible between laboratories [194].

Proteins are not static components of cells forming only binary interactions, but can also form indirect interactions within a protein complex or with nearby proteins [195]. The BioID method for the proximity labeling of interacting proteins is based on the ability of a modified biotin ligase (BirA*) to covalently attach a biotin on the neighboring proteins’ lysine residues upon biotin supplementation in the growth media [195]. A workflow for BioID sample analysis resembles AP-MS: the protein-of-interest (bait) is tagged with the BirA*, affinity-purified with streptavidin beads, and analyzed with LC-MS. Endogenous biotinylation in mammalian cells is mainly limited to carboxylases, thus providing a rather specific way to label close proximity interactors in living cells and to detect transient and weak interactions with a reasonable background [195,196]. It is noteworthy that the labeling radius is about 10 nm [197], indicating that spatiotemporally similar proteins will be labeled. Biotin will remain attached to the protein even though the BirA*-containing protein has moved further away, providing a molecular picture of the proteomics landscape in the cell.

### 4.2. Interactions of CDKs

CDKs interact with cyclins, a family of regulatory proteins, to form active CDK–cyclin complexes that regulate cell cycle progression [198]. Cyclins modulate CDK activity by promoting their catalytic activity and determining their substrate specificity [199]. CDK–cyclin complexes are regulated by CDK inhibitors (CKIs), which bind to CDKs and inhibit their activity, thus ensuring proper cell cycle control [200]. CDKs can also interact with various other proteins, such as transcription factors and chromatin remodeling enzymes, to regulate gene expression and cellular processes beyond cell cycle control [201].

### 4.3. Interactions of MAPKs

MAPKs are involved in highly conserved signaling cascades known as the MAPK pathways [202]. These pathways consist of a series of protein–protein interactions and phosphorylation events involving three classes of kinases: MAP kinase kinase kinases (MAP3Ks), MAP kinase kinases (MAP2Ks), and MAPKs [203]. MAP3Ks are activated in response to extracellular stimuli and phosphorylate MAP2Ks, which in turn phosphorylate and activate MAPKs. Activated MAPKs can then phosphorylate and regulate various downstream substrates, such as transcription factors, to modulate gene expression and cellular responses [202].

### 4.4. Interactions of GSKs

GSK-3 is known to interact with numerous proteins and is involved in various signaling pathways, such as the Wnt, insulin, and PI3K/Akt pathways [69,170]. GSK-3 can phosphorylate and modulate the activity of key proteins in these pathways, such as β-catenin in the Wnt pathway, glycogen synthase in the insulin pathway, and Mdm2 in the PI3K/Akt pathway [69]. GSK-3 also interacts with proteins involved in cell cycle regulation, apoptosis, and cellular differentiation, such as cyclin D1, c-Myc, and Bcl-2 family members [69].

### 4.5. Interactions of CLKs

CLKs interact with and phosphorylate serine/arginine-rich (SR) proteins, which are essential components of the spliceosome and play a critical role in the regulation of pre-mRNA splicing [204]. The CLK-mediated phosphorylation of SR proteins affects their localization, stability, and interaction with other spliceosome components, thereby modulating alternative splicing events [161,162]. The dysregulation of CLK-mediated alternative splicing has been implicated in various diseases, including cancer [160].

Further research is needed to identify additional interaction partners of CMGC kinases and elucidate their roles in the regulation of cellular signaling pathways. This knowledge will not only enhance our understanding of the molecular basis of CMGC kinase-mediated cellular processes but also facilitate the development of targeted therapeutic strategies for diseases associated with CMGC kinase dysfunction.

## 5. Assessment of Kinase Activity

As the function of protein kinases is to transfer a phosphate group, Western blotting with antibodies against a substrate’s phosphoserine, -threonine or -tyrosine is the most common and widely used method to indirectly identify kinase activity. This approach is very labor-intensive, highly dependent on the specificity and sensitivity of the available antibodies, and not directly comparable with high-throughput approaches.

A more sensitive, direct approach to quantitatively measure kinase activity in vitro is to measure the incorporation of radiolabeled phosphate from γ-^33^P ATP into the substrate [205]. A prerequisite to the solution-based in vitro kinase assays is a purified kinase, which in most cases is compared to the kinase dead mutant that abolishes ATP binding. An extensive study of 84 pairs of wild-type kinases and their kinase-dead counterparts [9] confirmed, using in vitro kinase assays, that indeed the mutated “VAIK” or “HRD” motif is essential for kinase activity. The lysine in the “VAIK” motif is among the most conserved amino acids in all kinases as it is a central catalyzer of the phosphate transfer [5].

Functional protein microarrays are an efficient ‘omics’-scale tool to detect kinase-substrate interactions [206,207]. The commercially available protein microarrays have thousands of purified proteins spotted onto a glass slide, providing an efficient means to identify the phosphorylation of candidate substrates for the kinases of interest when incubated together with radiolabeled ATP [208]. To date, the only large-scale study utilizing protein microarrays in kinase substrate identification has been performed with yeast protein kinases [209], although human protein microarrays are nowadays commercially available.

## 6. Conclusions

CMGC kinases play crucial roles in cellular signaling pathways, such as cell cycle regulation, proliferation, differentiation, apoptosis, and gene expression regulation [69,198,202,204]. Aberrant activation and dysregulation of these kinases have been implicated in cancer development and progression, highlighting their importance in cancer biology [162,201,203].

The therapeutic potential of targeting CMGC kinases has been demonstrated by the development and clinical success of kinase inhibitors, such as CDK4/6 inhibitors (palbociclib, ribociclib, and abemaciclib) for hormone-receptor-positive, HER2-negative, advanced or metastatic breast cancer [165,166,167], and BRAF and MEK inhibitors (vemurafenib, dabrafenib, trametinib, and cobimetinib) for metastatic melanoma-harboring BRAF mutations [132,133]. Furthermore, ongoing research efforts aim to develop novel inhibitors targeting GSKs and CLKs and explore their therapeutic potential in cancer and other diseases [171,210,211].

However, several challenges remain in targeting CMGC kinases for cancer therapy, including resistance to kinase inhibitors, off-target effects, and the need for better patient stratification [173,174,177,178]. To overcome these challenges, future research should focus on developing next-generation inhibitors with improved target selectivity, combination therapies targeting multiple signaling pathways, sequential treatment strategies, and exploiting protein–protein interactions involving CMGC kinases [175,176,180]. Additionally, identifying synthetic lethal interactions involving CMGC kinases and using non-invasive techniques, such as liquid biopsies, can facilitate the real-time monitoring of treatment response and early detection of resistance [179,182].

In summary, CMGC kinases are important in cancer biology, and targeting these kinases holds great therapeutic potential. Despite the challenges associated with their inhibition, ongoing research efforts and the development of novel therapeutic strategies promise to improve the efficacy and specificity of CMGC-kinase-targeted therapies. As our understanding of the molecular mechanisms and protein–protein interactions involving CMGC kinases expands, so too will the opportunities for the development of more selective and effective therapeutic strategies for cancer treatment.

## Figures and Tables

**Figure 1 cancers-15-03838-f001:**
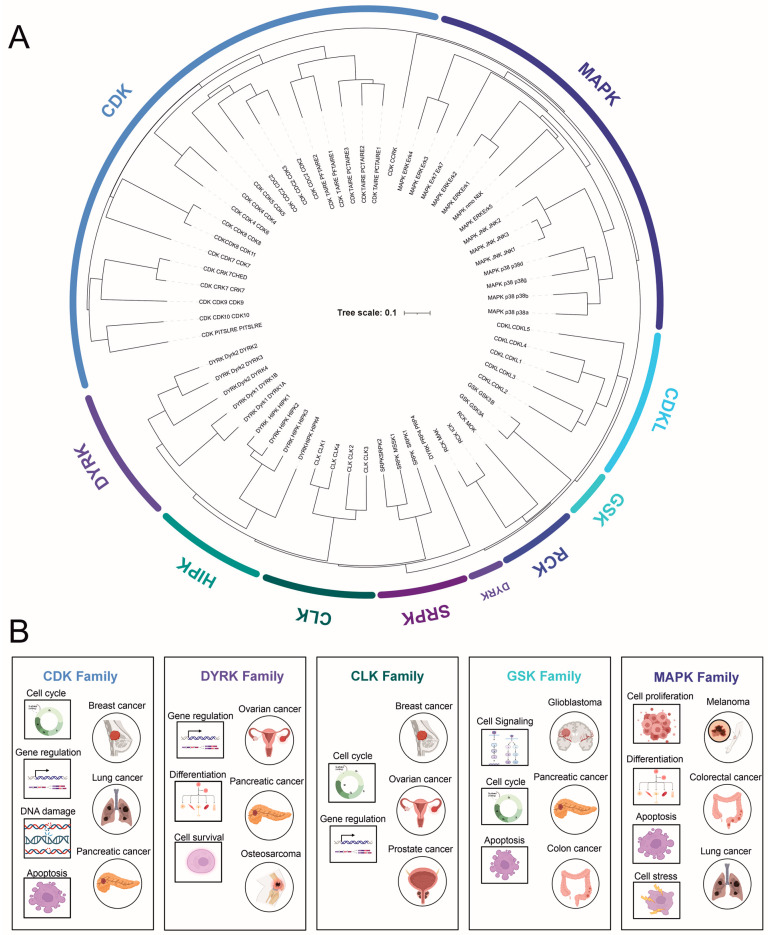
(**A**) A schematic diagram illustrating the taxonomy of the eight subfamilies of CMGC kinases based on their kinase domain sequence similarity. (**B**) CMGC kinase subfamily dysregulation in most common cancer types with the prevalent deregulated mechanisms involved in cancer progression. Images created with BioRender.com (accessed on 11 July 2023).

**Figure 2 cancers-15-03838-f002:**
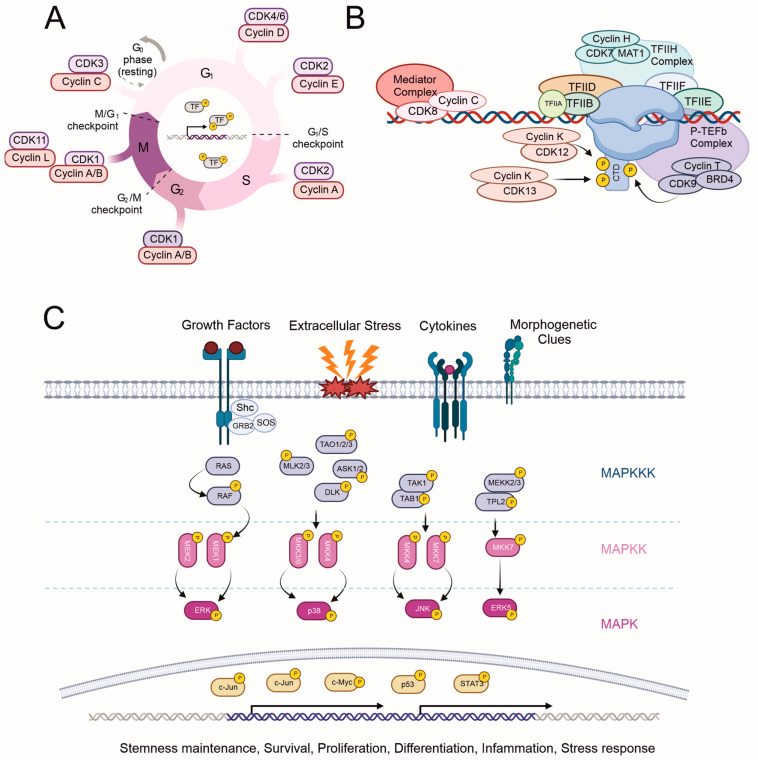
CMGC kinase signaling pathways in health and disease. A graphical representation of the major signaling pathways involving the two most studied families, (**A**,**B**) the CDK and (**C**) MAPK families, and their roles in promoting cancer hallmarks (e.g., cell proliferation, survival, migration, angiogenesis). Images created with BioRender.com (accessed on 16 July 2023).

**Figure 3 cancers-15-03838-f003:**
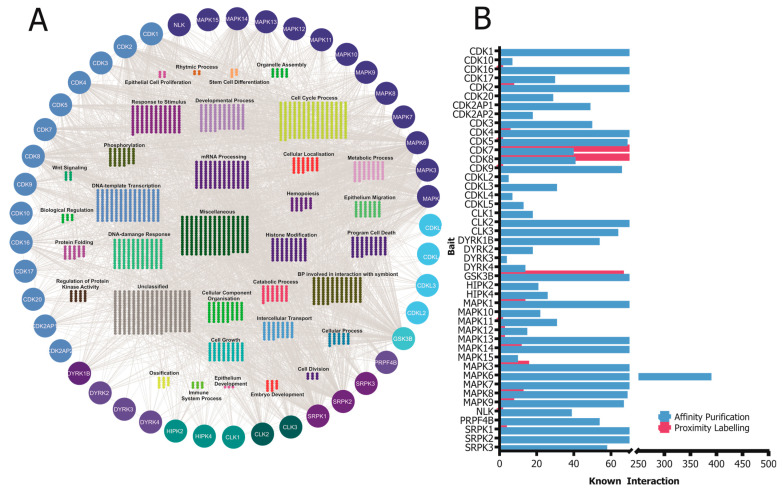
(**A**) A network of known high-confidence protein–protein interactions of CMGC kinases obtained from the IntAct Molecular Interaction Database are grouped according to their simplified GO process annotation. (**B**) Distribution of the number of known interactors of the CMGC kinases based on the affinity purification and proximity labeling detection method. The visualization of the protein–protein interaction (PPI) network was generated by importing the publicly available interaction dataset from https://www.ebi.ac.uk/intact (accessed on 8 May 2023) protein interaction data into Cytoscape v3.9.

**Table 1 cancers-15-03838-t001:** Prevalence and pattern of CMGC kinase subfamily dysregulation in cancer.

Kinase Family	Deregulation Mechanism	Examples of Cancer Types
CDKs	Mutations, amplifications, deletions, and altered expression levels	Breast cancer [127,128,129], lung cancer [130], pancreatic cancer [131]
MAPKs	Mutations in pathway components (e.g., RAS, RAF)	Melanoma [132,133,134], colorectal cancer [135], lung cancer [48,136]
DYRK	Overexpression	Pancreatic [137,138], ovarian cancer [139,140], osteosarcoma [141], rhabdomyosarcoma [142]
GSKs	Altered expression levels, post-translational modifications	Glioblastoma [143], pancreatic cancer [144], colon cancer [145]
CLKs	Overexpression, cancer-associated splicing alterations	Breast cancer [146], prostate cancer [147], ovarian cancer [148]

**Table 2 cancers-15-03838-t002:** Aberrant activation of CMGC kinases in cancer. N/A: information not available.

Kinase Class	Inhibitor Name	FDA Approval Status	Approved Indications	Key Clinical Benefits	Ongoing Research Areas
CDK	Palbociclib	Approved (2015)	HR-positive, HER2-negative advanced/metastatic breast cancer	Improved progression-free survival and overall response rates	Potential in other cancer types, combination therapies
CDK	Ribociclib	Approved (2017)	HR-positive, HER2-negative advanced/metastatic breast cancer	Improved progression-free survival and overall response rates	Potential in other cancer types, combination therapies
CDK	Abemaciclib	Approved (2017)	HR-positive, HER2-negative advanced/metastatic breast cancer	Improved progression-free survival and overall response rates	Potential in other cancer types, combination therapies
MAPK (BRAF)	Vemurafenib	Approved (2011)	Metastatic melanoma with BRAF mutations	Effective in BRAF-mutated melanoma, improved response rates	Targeting other components of MAPK pathways
MAPK (BRAF)	Dabrafenib	Approved (2013)	Metastatic melanoma with BRAF mutations	Effective in BRAF-mutated melanoma, improved response rates	Targeting other components of MAPK pathways
MAPK (MEK)	Trametinib	Approved (2013)	Metastatic melanoma with BRAF mutations	Improved outcomes when combined with BRAF inhibitors	Targeting other components of MAPK pathways
MAPK (MEK)	Cobimetinib	Approved (2015)	Metastatic melanoma with BRAF mutations	Improved outcomes when combined with BRAF inhibitors	Targeting other components of MAPK pathways
GSK	Tideglusib	Not Approved	N/A	Promising preclinical results in glioblastoma, pancreatic cancer	Further development, clinical evaluation in cancer treatment
CLK	TG-003	Not Approved	N/A	Modulates alternative splicing, reduces tumor cell viability (preclinical)	Therapeutic potential in cancer, diseases with aberrant splicing

**Table 3 cancers-15-03838-t003:** Potential combination therapies involving CMGC kinase inhibitors (https://clinicaltrials.gov/, accessed on 8 May 2023).

CMGC Kinase Inhibitor	Combination Agent	Rationale	Synergistic Effects	Clinical Trial Status
Palbociclib	Immune checkpoint inhibitors	Enhance antitumor immune response	Improved response rates	Ongoing clinical trials (NCT00141297)
Ribociclib	Angiogenesis inhibitors	Block tumor vascularization and growth	Enhanced tumor growth inhibition	Preclinical studies (NCT03285412)
Abemaciclib	Other kinase inhibitors	Target multiple signaling pathways simultaneously	Increased cell death, decreased proliferation	Ongoing clinical trials (NCT02057133)
Vemurafenib + Trametinib	Immune checkpoint inhibitors	Enhance antitumor immune response in combination with MAPK pathway inhibition	Improved response rates, prolonged survival	Ongoing clinical trials (NCT01597908)

## Data Availability

The publicly available interaction dataset from https://www.ebi.ac.uk/intact (accessed on 8 May 2023) was used to generate the protein–protein interaction landscape of CMGC kinases which is visualized in Figure 3.

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
