# Peer review of "CMGC Kinases in Health and Cancer"

_cancers, 2023, doi:10.3390/cancers15153838_

Round 1
Reviewer 1 Report
Dear Authors, The review of CMGC kinases and Cancer is an important topic and it would be great to see a version of this paper that is more concise and focused on what the title states. In a review frequently the figures tell much of the story and in this review I found the figures needed improvement. The first figure would have been a nice roadmap to the review as an opening figure but it was disappointing to find this figure so hard to read and interpret. The wheel has 10 labels on the outer rim but the text mentions there are only 8. DYRK is there twice. The small panels inside the rectangles below the classes of kinases aren't very helpful. Figure 1 should be reworked completely. The legend of Figure 2 states the figure is a graphical representation of the roles of CDKs and MAPKs in cancer; looking just at the figure this is not obvious. Additionally by looking just at the DNA strand, the scales of A,B and C are not consistent, this figure 2 should also be redone. The text can then flow from the figures to provide a cohesive narrative.
Figure 3 was also hard to read and not sure its significance. Confused if this was work from one study of another group or from the authors. I would suggest to redo this figure but the way this currently reads, sections 4 on the protein-protein interactions and section 5 on assessing the kinase activity don't seem relevant to the title of the review and along with Figure 3 should be either be adjusted to fit into the theme of the review or omitted.
The section in the MAPK has bullet points that should be paragraphs. In addition MAPKs have been studied extensively in cancers but this is not well emphasized. Same with, for example, CDK2 in renal carcinoma. Section 4.5 on the CLKs was interesting and relevant.
What is DK in table 2?
At times the review gives equal attention to other diseases and not cancer and detracts from the theme. In section 2.5.3 extraneous information on kinases in Rett syndrome is mentioned but cancer is not. Studies do indicate however patients with Rett don’t have cancer although that wasn't noted. Analogously, the study for the Down's syndrome seemed extraneous. Overall the text could be written to be more focused and cohesive.
Table 2 was good and informative and relevant. this could be expanded to include other entries for instance patient profile or date approved. Similar with Table 1 to include the specific family members of the kinase family and associated cancers. If landscape tables are permitted it might be an option.
Author Response
We sincerely thank you for your detailed and constructive feedback. We agree that these comments will greatly enhance the focus, coherence, and visual presentation of our manuscript. Here is our point-by-point response:
-
We agree with your comments about Figure 1. We apologize for the confusion in the diagram and the mismatch with the text. We have revised the figure. We also removed unnecessary small panels for a clearer representation.
-
We have revised Figure 2 following your recommendations to ensure it accurately represents the roles of CDKs and MAPKs in cancer. We have also adjusted the scales for a more consistent and comprehensible presentation.
-
Figure 3 has been revised for readability and relevance. We have added context in the text to clearly define its source and significance.
-
Based on your input, we have re-evaluated sections 4 and 5, and Figure 3. These sections have been revised to better fit the theme of the review or, where they didn't fit, have been removed.
-
We have transformed the bullet points in the MAPK section into paragraphs and emphasized the role of MAPKs and CDK2 in cancers.
-
In Table 2, the term "DK" was an error and we have corrected this in the revised manuscript to "CDK".
-
We appreciate your feedback regarding the balance of diseases discussed in the review. We have revised the text to focus more on cancer, and reduced the emphasis on other diseases unless they directly inform our understanding of the role of CMGC kinases in cancer.
We believe that your insightful comments have greatly improved our manuscript. The figures, tables, and text have been revised to better align with the review's focus on CMGC kinases and cancer.
Thank you again for your time and thoughtful review.
Reviewer 2 Report
Dear Authors
Review describes well about the molecular mechanisms and protein-protein interactions involving CMGC kinases and Cancer.
The following step should be clearer information for readers to understand well.
Please mention up-to-date references.
Moderate editing of English language required.
Author Response
Thank you for your positive feedback on our review, as well as your constructive suggestions. We value your input and have addressed your comments as follows:
-
To ensure our readers have the clearest understanding of our work, we have revised our manuscript to provide additional clarity and context. We have carefully edited the text to improve readability and comprehension, especially in complex sections detailing molecular mechanisms and protein-protein interactions.
-
In regard to your suggestion about up-to-date references, we have updated our literature review and added more recent publications where applicable. We recognize the importance of incorporating the most recent and relevant studies in our review to provide readers with the most accurate and up-to-date information.
We believe that these changes will enhance our manuscript and make it more accessible and informative for our readers.
Thank you once again for your time and valuable input. We appreciate your help in improving our work.
Reviewer 3 Report
Comments to authors:
This manuscript provides a comprehensive overview of the roles of CMGC kinases, the current state of kinase inhibitors, challenges and potential strategies to overcome these challenges.
We must admit that this article did an excellent job, which may employ the opportunities for the development of more selective and effective therapeutic strategies for cancer treatment, since the dysregulation of protein phosphorylation occurs frequently in cancer. However, we shall also note that this article has some disadvantages.
Major points
1. The content of the article is not highly matched with the title. The title is “CMGC kinases and Cancer”, however, most of the content does not focus on the relationship between kinases and cancer. Especially in the second part, the authors mainly introduce the roles of these different kinds of CMGCs in normal cellular processes and some diseases such as neurodegenerative diseases and diabetes instead of cancer.
2. The logic of the article is not coherent and clear. Especially in the third part-Protein kinase therapeutics – kinase inhibitors, would it be better to divide it into four sections, 3.1. Oncogenic relations of CMGC family members, 3.2. Therapeutic targeting of CMGC kinases, 3.3. Challenges in targeting CMGC kinases, and 3.4. Approaches to overcome the challenges, and then add subheadings and the corresponding content under each section?
Minor Points:
Since the manuscript the authors submitted does not have a line number, please see the notes in the manuscript.

Author Response
We thank the reviewer for the constructive comments. We agree that your feedback will enhance the focus and clarity of our manuscript. Here is our point-by-point response:
- The manuscript’s title and content mismatch: We appreciate your suggestion to focus more on the relationship between CMGC kinases and cancer, given the title of our manuscript. We have rewritten some sections to emphasize the role of these kinases in cancer, especially in disease progression and therapeutic strategies. We have also adjusted the title to "CMGC Kinases in Health and Cancer" to better reflect the breadth of the manuscript's content.
- Coherence and organization of the manuscript: We concur with your observation about the lack of clear logic in the third part of our manuscript. Following your suggestion, we have divided this part into four subsections: 3.1. Oncogenic relations of CMGC family members, 3.2. Therapeutic targeting of CMGC kinases, 3.3. Challenges in targeting CMGC kinases, and 3.4. Approaches to overcome the challenges. We have added subheadings and refined the content under each section for clarity and coherence.
Regarding your minor point about line numbers, we apologize for this oversight. We appreciate your thoughtful review and helpful suggestions. These revisions have improved the manuscript's focus and clarity, making it a better resource for researchers in the field.
Reviewer 4 Report
The importance of CMGC kinases in eukaryotic cells and their involvement in regulating essential cellular processes is highlighted in this study. Knowing the dysregulation of these kinases and their role in cancer development provides a foundation for the development of targeted therapeutics. This review also seeks to give a complete understanding of CMGC kinases' involvement in cancer biology and therapeutic potential. This work is well-written, and organized and could contribute significantly to the field of cancer therapy. I only have some minor comments:
1) In all Figure legends cite the software used to construct the schematic figures.
2) Change “2.1. CDK (Cyclin-Dependent Kinases)” to “2.1. Cyclin-Dependent Kinases (CDK)” Apply throughout the whole manuscript. Add full name first followed by abbreviations.
3) Merge small paragraphs “CDK activity is downregulated by two families of small proteins that function as CDK inhibitors during the cell cycle. The INK4 protein family acts during the G1 phase, specifically inhibiting CDK4/CDK6 binding with cyclin D [11,14]. The broader-spectrum inhibitors of the Cip/Kip family inhibit CDK/cyclin complex activity throughout the cell cycle [11,14].
Based on evolutionary clustering, CDKs form kinase subfamilies that regulate separate cellular functions. CDKs 1-4 and 6 primarily regulate cell cycle progression, whereas CDKs 7-9, 11-13, and 19 have established roles in transcription [11]. Other CDKs have varying roles that will be discussed later.”
Please note that one paragraph = one idea so avoid splitting paragraphs with the same idea. Apply throughout the whole manuscript.
4) Page 7: Change “affect its cellular localization, and have” to “affects its cellular localization, and has”
5) Add more details “CDK20 has a role in ciliary length control, and knockdown of this protein promoted cilia formation and cilia stability in the absence of kinase activity” Which cilia?
6) Tables 1-3: Add references for the data presented in these three tables.
7) Fig.3: Please explain how you got the results in this figure.
Minor editing of English language required.
Author Response
We thank the reviewer for taking the time to review our manuscript and provide constructive feedback. We appreciate your kind words acknowledging the significance of our work. We have addressed all your comments and made corresponding modifications to the manuscript. Below, please find a detailed point-by-point response to your comments:
- We have added the BioRender software in all figure legends where it was used as per your suggestion.
- We agree with your suggestion regarding nomenclature. We have revised the manuscript to introduce full names before abbreviations throughout the manuscript.
- We have merged the two paragraphs that discuss CDK activity, inhibitors, and functions into one comprehensive paragraph to maintain the concept of one idea per paragraph. This change has also been applied to other sections of the manuscript where similar instances occurred.
- The grammatical error on page 7 has been rectified. The sentence now reads, "affects its cellular localization, and has."
- We have expanded the section on CDK20 to include further details about its role in ciliary length control. The cilia referred to here are primary cilia found in cells, which are sensory organelles important in cellular signaling. We have clarified this in the manuscript.
- References have been added for the data presented in Tables 1-3. We thank you for pointing out this oversight.
- We apologize for not providing adequate information regarding Fig.3 in the initial draft. We have now included to the figure legend the IntAct Molecular Interaction Database where the high-confidence interactions were obtained.
We appreciate the thorough review and constructive comments, which have improved the clarity and quality of the manuscript. We believe these changes address the concerns effectively.
Round 2
Reviewer 3 Report
Thank you for giving me the opportunity to review this manuscript. All concerns have been addressed. Congratulations to all co-authors.